# On the intermittency of orographic gravity wave hotspots and its importance for middle atmosphere dynamics

Ales Kuchar<sup>1</sup>, Petr Sacha<sup>2,3</sup>, Roland Eichinger<sup>4,5</sup>, Christoph Jacobi<sup>1</sup>, Petr Pisoft<sup>2</sup>, and Harald E. Rieder<sup>3</sup> <sup>1</sup>Institute for Meteorology, Universität Leipzig, Stephanstr. 3, 04103 Leipzig, Germany <sup>2</sup>Department of Atmospheric Physics, Faculty of Mathematics and Physics, Charles University, V Holesovickach 2, 180 00 Prague 8, Czech Republic <sup>3</sup>Institute of Meteorology and Climatology, University of Natural Resources and Life Sciences, Vienna (BOKU), Gregor-Mendel-Strasse 33, 1180 Vienna, Austria <sup>4</sup>Meteorological Institute, Ludwig-Maximilians-University (LMU), Munich, Germany <sup>5</sup>Deutsches Zentrum für Luft- und Raumfahrt (DLR), Institut für Physik der Atmosphäre, Oberpfaffenhofen, Germany **Correspondence:** ales.kuchar@uni-leipzig.de

Abstract. When orographic gravity waves (OGWs) break, they dissipate their momentum and energy and thereby influence the thermal and dynamical structure of the atmosphere. This OGW forcing mainly takes place in the middle atmosphere. It is zonally asymmetric and strongly intermittent. So-called 'OGW hotspot regions' have been shown to exert a large impact on the total wave forcing, in particular in the lower stratosphere (LS). Motivated by this we investigate the asymmetrical

- distribution of the three-dimensional OGW drag (OGWD) for selected hotspot regions in the specified dynamics simulation of the chemistry-climate model CMAM (Canadian Middle Atmosphere Model) for the period 1979-2010. As an evaluation, we first compare zonal mean OGW fluxes and GW drag (GWD) of the model simulation with observations and reanalyses in the northern hemisphere. We find an overestimation of GW momentum fluxes and GWD in the model's LS, presumably attributable to the GW parameterizations which are tuned to correctly represent the dynamics of the southern hemisphere. In
- the following, we define three hotspot regions which are of particular interest for OGW studies, namely the Himalayas, the Rocky Mountains and East Asia. The GW drags in these hotspot regions emerge as strongly intermittent, a result that can also quantitatively be corroborated with observational studies. Moreover, a peak-detection algorithm is applied to capture the intermittent and zonally asymmetric character of OGWs breaking in the LS and to assess composites for the three hotspot regions. This shows that LS peak OGW events can have opposing effects on the upper stratosphere and mesosphere depending
- on the hotspot region. Our analysis constitutes a new method for studying the intermittency of OGWs, thereby facilitating a new possibility to assess the effect of particular OGW hotspot regions on middle atmospheric dynamics.

# 1 Introduction

Internal gravity waves (GWs) are a naturally occurring and ubiquitous phenomenon with large impact on the atmosphere's thermal and dynamical structure (Andrews and McIntyre, 1987; Fritts and Alexander, 2003). While the Brewer-Dobson cir-

20 culation is believed to be driven mainly by Rossby waves, the mesospheric global circulation from the summer to the winter pole is dominated by GWs (Plumb, 2002; Alexander, 2013). GWs contribute to mesospheric cooling which often accompanies

Sudden Stratospheric Warmings (SSWs, Stephan et al., 2020) and they may also play an important role in vortex preconditioning (Albers and Birner, 2014).

- In the current generation of general circulation models (GCMs), the resolution is usually too coarse to simulate GWs directly, requiring that the majority of their spectrum must be parameterized. Usually, two parameterization schemes are employed to distinguish between orographic (OGWs) and non-orographic GWs (NGWs). All GW parameterizations employ various degrees of simplification of GW sourcing, propagation and dissipation processes and contain certain tunable parameters that are only poorly constrained by observations. The performance of the schemes is commonly evaluated through comparison of zonal mean climatologies of GCMs and observations (Geller et al., 2013).
- From observations, GWs are known to be distributed spatially asymmetric around the globe in so-called hotspots (e.g. Hoffmann et al., 2013). The asymmetry of the spatial distribution of the total GW drag (GWD) resulting from the two parameterizations is well represented by the OGW parameterizations (Šácha et al., 2018). OGWD hotpots are associated with well-known topographic structures such as the Andes and the Antarctic Peninsula in the southern hemisphere (SH), and the Rocky Mountains, the Scandinavian range and the Himalayas in the northern hemisphere (NH). These structures produce zon-
- ally asymmetric and interanually-variable torques, which significantly contribute to the total drag, emerging already as low as in the lower stratosphere (LS, Šácha et al., 2019).

Recent observational studies have shown that GW activity is highly intermittent (e.g. Hertzog et al., 2012; Wright et al., 2013) in terms of large amplitude wave-packets. In the present study, we focus on the valve layers in the LS (Kruse et al., 2016; Bramberger et al., 2017), where weak or zero horizontal winds between the subtropical jet and the polar night jet allow OGWs

- to break and deposit the momentum (and energy). As a first study of this kind, we will investigate the short-term variability of the three-dimensional (3D) OGWD in a GCM simulation. For this, we explore outputs of a transient CMAM (Canadian Middle Atmosphere Model, McLandress et al., 2013) simulation with specified dynamics. Our study starts with a model description, a short review of its evaluation and a brief description of other datasets used in the study in Section 2.1. In Section 2.2, we describe the methodology allowing to attribute the intermittency of parameterized OGWs, which leads to short (on a daily
- timescale) and strong bursts of localized wave forcing in the lower stratosphere. The simulated OGWD is compared with recent observational datasets in a traditional zonal mean monthly mean manner in Section 3.1. In Section 3.2 we present a statistical analysis of the OGWD within hotspots and analyze its intermittency. Finally, we present first results of a new method for studying the impact of spatiotemporally intermittent OGWD in Section 3.3, and end with concluding remarks in Section 4.

# 2 Data and methodology

## 50 2.1 Description of model and observations

The study is based on a simulation performed with the CMAM simulation (McLandress et al., 2013). CMAM is a chemistryclimate model with 71 vertical levels spanning from the surface up to  $7 \cdot 10^{-4}$  hPa (about 100 km) with variable vertical resolution. It uses a triangular spectral truncation of T47, but the physical parameterizations are performed on a 3.75° horizontal grid. We selected a transient model simulation covering the period 1979–2010 with specified dynamics up to 1 hPa (referred

- hereinafter as CMAM-sd). This means that Newtonian relaxation ("nudging") on spatial scales of <T21 to the 6-hourly horizontal winds and temperature time series from ERA Interim (Dee et al., 2011) is applied. For further details about the nudging we refer the reader to McLandress et al. (2014). The upper stratospheric discontinuities in the reanalysis data emerging in 1979, 1985, and 1998 have been removed from the model data using the procedure described in McLandress et al. (2014). CMAM-sd has been chosen for our analysis due to the freely accessible 6-hourly model data including 3D GW diagnostics, which to our</p>
- knowledge is currently unique in model data repositories. Moreover, CMAM is widely known for its realistic representation of middle atmospheric dynamics and has extensively been evaluated (see below).

In CMAM, OGWs and NGWs are parameterized using the schemes of Scinocca and McFarlane (2000) and Scinocca (2003), respectively. While McLandress et al. (2013) extensively detail both parameterization configurations, a brief outline of the parameterizations is given below. The OGW scheme launches two vertically propagating zero-phase-speed waves with orien-

- tation and magnitude depending on the near-surface static stability, wind speed and direction relative to the subgrid topography (anisotropic effects). Two tunable parameters exist in this parameterization scheme: the integrated radial dependence of the pressure drag (G(y) = 0.65) scaling the total vertical flux of horizontal momentum and the inverse critical Froude number (Fr<sub>crit</sub> = 0.375) determining the breaking height that have been tuned to reduce warm temperature biases in the SH climatology (Scinocca et al., 2008). The NGWD scheme considers a spectrum of non-zero phase speed GWs propagating horizontally
- <sup>70</sup> into four cardinal directions at the fixed launch level ( $\sim 125 \,hPa$ ) with a pre-defined launch flux ( $\sim 10^{-4} \,Pa$ ). These parameters are tuned to exert a reasonable drag in the upper stratosphere and mesosphere (McLandress et al., 2013).

CMAM-sd has been vastly evaluated by means of comparisons with observations (e.g. Shepherd et al., 2014). Climatology of zonal winds and temperatures in the lower to middle stratosphere in CMAM-sd have been found to be consistent with reanalyses and observations, although some local biases have been identified higher in the upper stratosphere and mesosphere (Shepherd

et al., 2014; Pendlebury et al., 2015; Kuilman et al., 2017).

In the first section of the results, we compare the OGWD of the CMAM-sd simulation with the most recent generation of NASA's reanalysis MERRA2 (Modern Era Reanalysis for Research and Applications-2) version of the Goddard Earth Observing System-5 (GEOS-5, Molod et al., 2015), the Japanese 55-year Reanalysis (JRA55, Ebita et al., 2011) and with the observation-based GW climatology dataset GRACILE (Ern et al., 2018). MERRA2 uses both orographic (McFarlane,

- 1987) and non-orographic (Garcia and Boville, 1994) wave parameterizations (see details in Fujiwara et al., 2017), while JRA55 uses an OGW parameterization only (Iwasaki et al., 1989). GRACILE has been compiled with data from the SABER instrument on NASA's TIMED (Thermosphere Ionosphere Mesosphere Energetics Dynamics) satellite together with data from HIRDLS (High Resolution Dynamics Limb sounder) aboard NASA's Aura satellite. Note, SABER data are not assimilated in MERRA2. The GRACILE data set is suitable for comparison with GW distributions in global models either with parameterized
- or resolved GWs. GW momentum fluxes from SABER and HIRDLS have been used previously for comparison with selected climate models and radiosonde observation by Geller et al. (2013).

**Figure 1.** Ratio of zonally averaged OGWD in zonal direction (units: %) to the total wave forcing (resolved waves represented by EPFD + OGWD + NGWD) for the climatological average of the period 1979-2010. The black contour represents 50% contribution of OGWD.

# 2.2 Construction of hotspot composites

Figure 1 shows the boreal winter (DJF) average of the zonal-mean OGWD contribution to the total (OGWD+NGWD+resolved wave drag represented by Eliassen-Palm flux divergence (EPFD)) zonal mean wave drag in the NH in CMAM-sd. Here two

- regions emerge in the middle atmosphere where the OGWD dominates the net drag, namely the lower mesosphere and the LS. In the lower mesosphere, OGWD controls most of the net drag between 40 and 75°N in all months with the exception of the boreal summer months (not shown). In the LS between 25 and 50°N, OGWD constitutes the majority of the net drag during boreal winter and adjacent spring and autumn months. The region of the LS OGWD maximum in the extratropics at 70 hPa starts at the upper flank of the subtropical jet and extends into the area of weak winds below the polar night jet. According to
- theoretical considerations postulated in Teixeira (2014) or following observational evidence from lidar measurements (Ehard et al., 2017), these areas, known as the valve layers (e.g., Kruse et al., 2016; Bramberger et al., 2017), are regions where weak or zero horizontal winds provide critical levels for OGWs. There, they break and deposit horizontal momentum. The dominance of the zonal mean OGWD forcing in the NH LS net drag emerges also as a robust feature in the free running simulations with global (chemistry) climate models (including CMAM, Šácha et al., 2019; Okamoto et al., 2011; Dietmüller et al., 2018). In
- reanalyses, GWD constitutes about half of the net forcing in the LS (Albers and Birner, 2014; Abalos et al., 2015; Sato and Hirano, 2019).

The dominant OGWD in the LS at 70 hPa is mostly distributed into hotspots connected with regions of distinct topography. In our analysis, we focus on the hotspots highlighted by the colored boxes in Fig. 2. The amber, purple and green boxes represent the Himalaya (HI, 70-102.5°E and 20-40°N), East Asia (EA, 110-145°E and 30-48°N) and West America (WA,

235-257.5°E and 27.5-52°N) hotspots, respectively. The HI and WA hotspot areas have been defined based on the mountain range locations. The definition of the EA hotspot is not that straightforward as it corresponds to a geographical location of multiple mountain ranges. Several studies have reported the importance of the EA region as a "vertical communicator" from the troposphere into the stratosphere (e.g. Nakamura et al., 2013; Cohen and Boos, 2017; White et al., 2018). Strong OGW activity in the LS in this region has also been shown in observations. Šácha et al. (2015) have highlighted peak GW activity