# Peer review of "On the intermittency of orographic gravity wave hotspots and its importance for middle atmosphere dynamics"

_Weather and Climate Dynamics, 2020_

## Referee Comment (RC1) · Anonymous Referee #1 · 10 Jun 2020

Based on model output of the GCM CMAM-sd for the years 1979-2010 the authors introduce a novel method to investigate the intermittent nature of parameterized orographic gravity wave forcing (OGWD). Three hotspot regions are investigated, the Himalayas, the Rocky Mountains, and East Asia. It is found that OGWD in the hotspot regions is very intermittent, as expected from previous simulations and observations. By means of a peak-detecting algorithm, a set of strong peak events is identified and investigated in more detail. Composite analysis shows that during peak events the effect on the upper stratosphere and mesosphere differs strongly between the different hotspot regions, and effects can even have opposite sign.

In summary, this is an interesting paper that provides new and unique information which is of interest to the readership of WCD. The paper is well written and the figures are of good quality. Therefore the paper should be published in WCD after addressing my major comment and my detailed comments.

Major comment:

My major concern is more of technical than of scientific nature. Part of the work has been published before in the first author's PhD thesis, and figures or parts of them are just reproduced. The PhD thesis, however, is not even listed among the references.

This concern can easily be resolved by referring properly to the first author's PhD thesis, see the "specific comments".

Further, I want to state clearly that the scientific value of the paper is not affected because the duplications occur mainly in the "Data and methodology" part of the paper, and the figures are definitely needed as an introduction to the intermittency analysis that is new and not contained in the PhD thesis.

Link to the PhD thesis:

https://dspace.cuni.cz/bitstream/handle/20.500.11956/102077/140068896.pdf

Detailed comments:

(1) p.2, l.30/31: This statement is too specific! Perhaps because also the reference Hoffmann et al., 2013 is too specific as it covers only orographic and convective hotspots and is very limited by the observational filter of the AIRS satellite instrument. In addition to the hotspots, there is also considerable background gravity wave activity, as can be seen from other gravity wave climatologies (for example, Ern et al., 2018), and also gravity waves emitted from other sources like jets and fronts (almost invisible for AIRS) can exhibit spatial asymmetries and can occur as hotspots. This becomes most obvious, for example, during sudden stratospheric warmings (see, for example Ern et al., 2016)

Ern, M., Trinh, Q. T., Kaufmann, M., Krisch, I., Preusse, P., Ungermann, J., Zhu, Y., Gille, J. C., Mlynczak, M. G., Russell III, J. M., Schwartz, M. J., and Riese, M.: Satellite observations of middle atmosphere gravity wave absolute momentum flux and of its vertical gradient during recent stratospheric warmings, Atmos. Chem. Phys., 16, 9983-10019, 2016.

Therefore I would suggest to modify the statement in l.30/31 as follows:

"From observations, GWs are known to be distributed spatially asymmetric around the globe in so-called hotspots (e.g. Hoffmann et al., 2013)."

->

"From observations, it is known that, in addition to relatively uniform background gravity wave activity, specific gravity sources like orography, convection, or jets and fronts, can occur as so-called hotspots and introduce strong spatial asymmetries around the globe (e.g. Hoffmann et al., 2013, Ern et al., 2016, 2018)."

(2) p.2, l.31/32: This statement is too unspecific! The statement may hold also for other models that do not use specified non-orographic gravity wave sources like convection, or jets and fronts. However, here and in Sacha et al. (2018) only CMAM-sd is investigated. Further, the step from the real observations that show also multiple features of non-orographic gravity waves to the parameterized gravity waves in CMAM-sd with a simple non-orographic source is too large. Suggestion:

resulting from the two parameterizations is well represented by the OGW parameterizations

->

resulting from the two parameterizations in CMAM-sd is mainly introduced by the OGW parameterization

(3) p5, l.113/114 How is area weighting performed? In terms of square-km areas

assigned to each grid point? (This information is important because grid points at lower latitudes would be weighted more strongly.)

(4) p.5, l.121: Not clear what "with minimum distance of 20 days" means. Has a detected peak event to be separated by more than 20 days from the next peak event? Please clarify!

(5) p.7, l.153: In Geller et al. (2013) CMAM-sd is not mentioned, but the main statement still holds. Perhaps just rewrite the first part of the sentence: "Comparison with Fig.2 in Geller et al. (2013) reveals that CMAM-sd ..."

(6) p.8, l.183: Considering the changed shape seen in Fig. S2 after revision, why would you think that it is rather missing OGWD, and not NGWD, if the shape changes from a double-peak to a single peak?

(7) About p.10, l.118/119 onward, and the supplement: please add some explanation! Strong intermittency in summer, in non-mountainous regions, or at higher altitudes does not necessarily indicate "strong OGWD", but could be the effect that OGWD occurs only very sporadically. In these cases, compared to the total GWD, OGWD could be just negligible.

(8) p.11, l.229/230 Intermittency lost by zonal averaging could also happen if other GW sources are more dominant than the OGWs in the hot spots.

(9) p.11, l.232: Please state more clearly that during winter OGWs can permanently find favorable propagation conditions and act more continuously. Therefore OGW intermittency is reduced in winter, but probably OGWD more relevant.

(10) p.11, l.234: About multiple mountain ranges - what do you think is the mechanism? Could it be that if multiple mountain ranges are present (and have different orientations) there are more often favorable OGW propagation conditions for at least one of the ranges, with the effect of OGWs acting more continuously than in other regions with only a single mountain range?

(11) p.12, l.269 please repeat the information that the considered peak events are mainly in the winter season when winds are usually eastward (positive) throughout the stratosphere and lower mesosphere.

(12) p.12, l.276: Would this correspond to positive zonal wind anomalies on zonal average?

(13) Fig.6 If you want to make the statement that the OGWD pdf distributions are log-normal, please add theoretical curves (the corresponding fits of log-normal distributions).

(14) In the caption of Fig.8, "p-values" are mentioned, but not explained. Please clarify: are these values significance values arising from t-tests?

(15) p.15, l.322: Please clarify that "minima" means "strong negative values".

(16) p.4, Fig.1 contains 3 panels that were published before in the PhD thesis Kuchar (2018), Fig. 3.4 there. At least a reference to the thesis should be included. Perhaps something like "Adapted from Kuchar (2018)."

(17) p.6, Fig.3 is almost exactly the same as in the PhD thesis Kuchar (2018), Fig. 3.7 there. Again, at least a reference to the thesis should be included.

(18) p.13, also Fig.8 is similar to one of the figures in the thesis (Fig. 3.12)

Technical comments:

p.2, l.51: CMAM simulation -> CMAM

p.5, l.121: beneath -> beyond

p.6, l142: satelite -> satellite

p.9, l.205: 6 -> -6

p.14, l.283: hotspot composites can have an impact -> hotspot composites indicate potential impacts

p14, l.288: have been -> has been

---

## Referee Comment (RC2) · Corwin Wright (Referee) · 21 Jun 2020

The authors use output from a specified-dynamics run of the Canadian Middle Atmosphere Model (CMAM) to investigate how its orographic gravity wave parameterisation captures the variability at short time and space scales ('intermittency') of gravity wave forcing. They focus on three northern-hemisphere wave 'hotspots', namely the Himalayas, Rocky Mountains and the generalised high topography of north-east Asia. They compare their results to intermittency in observational data from limb-sounding satellites and to previous studies. They conclude that (i) the assumption of vertically-propagating waves in parameterisations is realistic in the lower stratosphere; (ii) that

peak events can have opposing effects on the upper stratosphere and above depending on the specific region chosen, and (iii) that in the zonal mean positive OGWD anomalies contribute to mesospheric cooling.

The paper is clearly laid-out and well-written, and I have no major criticisms of the content or the presentation. In particular, the work is well-contextualised within the literature, the figures well-designed (with good use of colour to avoid significant issues for red/green colourblind readers), and the implications of their results are made clear. There are a few minor spelling and grammar errors, but these should be easily resolvable during the copy-editing stage. I support publication with at most minor revisions.

I have only one significant comment, which should not impede publication. This is that in Section 3.3 (Composite Analysis) they justify their choice to analyse the data at the regional rather than local scale with the statement that "Figs. 6 and 7 showed that spatial averaging inside the hotspots maintains the intermittent feature of the OGWD and only the information of the long tails of the distribution with extreme (and apparently very localized) drag values of $-20m/s/day$ is lost". I disagree with this statement, especially as relates to Figure 6 - to my eye, the form of the regional-scale distributon (blue bars) is quite distinct to that of the grid-scale distribution (coloured bars). I agree that the appropriate step to take in Section 3.3 is to analyse at the regional scale, and in particular that this is necessary to make the results useful for model developments - I just disagree with this justification, and think it would be better to admit that the distributions are a bit different and explain why their results are still meaningful despite this.

In addition to this I have a series of minor specific comments - most of these are requests for increased clarification in the text rather than errors.

L019: the paper comes in at quite a high level without much preamble, and in particular the mention of the BDC without any contextualisation might be hard for someone entering the field to follow. Would suggesting adding a brief description of what this to

soften it for e.g. a starting research student.

L031: you refer to the GWs as being spatially asymmetric, but don't say in what direction. From later in the paper I assume this is E/W rather than (e.g.) N/S, but you should mention this here.

L051: This sentence expands out to "The study is based on a simulation performed with the [Canadian Middle Atmosphere Model] simulation". Suggest rephrasing to be a bit less repetitious.

L053: how variable a vertical resolution? Is it high enough for your purposes in the regions you study? Suggest briefly clarifying this.

L055: you refer to "spatial scales of <T21". I assume by this you mean scales which are physically larger than T21, but this is not what the sentence says - it instead refers to (relatively) small-physical-scale features. Suggest rephrasing for clarity.

L064: you say the parameterisation launches "two vertically propagating zero-phase-speed waves" but not how often. Per timestep? Per day? Please clarify.

L071: you say the parameters are tuned to exert a "reasonable" drag. In what sense?

L083: you also need to clarify: (i) whether HIRDLS was assimilated in MERRA [I assume not] and (ii) whether HIRDLS or SABER were assimilated by JRA55.

L088: missing close-bracket.

L100: "GWD constitutes about half of the net forcing in the LS" - at the global, zonally-local, or regional scale?

Figure 2: I would strongly suggest extending this figure further south - the boxes you highlight are so close to the "horizon" that it is very difficult visually to work out the areas they cover. This is particularly the case for the Himalayas box - I have real trouble working out which parts of the mountain range it covers.

[Figure]

L115: You say that the meridional accelerations were not analysed at all. Are you certain that they have cannot make a significant contribution? For example, over the edge of Antarctica (a long way from your regions!), mountain-generated GWMFs often have significant meridional components up to 50% of the zonal component - is that definitely untrue for these Northern hemisphere regions? In particular, since the main ridge of the Himalayas is aligned at a slight angle to the E-W direction, I would perhaps expect some degree of meridional forcing from there, even if not the Rockies since they're so cleanly N-S.

L145: the large difference between HIRDLS/SABER and the model is in fact a lot less than it looks from the raw numbers, and definitely less in reality than the factor of two or more inferred here. This is due to two problems affecting limb-sounder GW measurements. Firstly, and most importantly, these satellites are making a 2d cut at a high angle through the 3d wave field, travelling primarily meridionally. As a result, if you are trying to measure zonally-oriented waves (which is implicit in your comparison), then these instruments will massively underestimate wave momentum fluxes, since they measure the PROJECTION of the horizontal wavelength in the ALONG-TRACK direction. This is discussed well by e.g. Ern et al (JGR, 2004) and discussed in slightly more detail in Wright et al (ACP, 2015), and this effect will massively low-bias the measured momentum fluxes. Secondly, and less importantly, spectral fitting methods such as those used to measure GWs in these data will inherently underestimate peak amplitudes, again skewing the data low. Indeed, I would expect the values in GRACILE to be closer to a lower bound than a true estimate of the actual MF. I'm not sure how to integrate this into your text, but a brief mention that these limb-sounder estimates are known to be significantly low-biased at a systematic level would be helpful.

L173: word choice is a little sticky here: a "negative [gravity-wave drag]" is, literally, a gravity-wave acceleration. Would suggest a slight rephrase to make absolutely clear what you mean.

Figure 4: it's quite tricky to follow the seasonal cycle in the GRACILE data due to the

[Figure]

scale. Maybe try a log-scale ordinate?

Figure 5: same comment for the NOWD line.

Figure 6 caption and Line 194: I think you have the colours of EA and WA described the wrong way around.

L207: "Surprisingly, the second most frequent OGWD value is a small positive drag for all hotspots". Is this that surprising? For example, +1 is a lot closer to -1 than -50 is, so it seems intuitive to me that a small positive value would occur more frequently than a large negative value given the form of the distribution. Would suggest removing the line.

Table 1: Why are the months numbered rather than named, given you discuss them by name? And why are SON and MAM included here but not JJA, when they haven't been in the rest of the paper.

Code/data availability statement: excellent, but the first two sentences are ungrammatical.

---

## Author Comment (AC2) · 24 Aug 2020

We thank the reviewer for his/her thoughtful comments and suggestions. Please see the attached supplement for our detailed responses.

Please also note the supplement to this comment:
https://wcd.copernicus.org/preprints/wcd-2020-21/wcd-2020-21-AC2-supplement.pdf
* * *

---

## Author Comment (AC3) · 24 Aug 2020

We thank Corwin Wright for his/her thoughtful comments and suggestions and the overall enthusiastic review. Please see the attached supplement for our detailed responses.

Please also note the supplement to this comment:
https://wcd.copernicus.org/preprints/wcd-2020-21/wcd-2020-21-AC3-supplement.pdf

---

## Author Response (AR2)

Dear co-editor and all reviewers,

we thank all reviewers for their additional comments. The remaining technical suggestions by reviewer 2 were addressed. Furthermore, I could not change the link to the cited PhD thesis since no other link to the English version of the webpage exists. Unfortunately, there are a few Czech expressions.

Best regards
Ales Kuchar